

# Multiple comparative metagenomics using multiset *k*-mer counting

Gaëtan Benoit[1], Pierre Peterlongo[1], Mahendra Mariadassou[2], Erwan Drezen[1,3], Sophie Schbath[2], Dominique Lavenier[1] and Claire Lemaitre[1]

[1] Inria Rennes Bretagne Atlantique - IRISA, GenScale team, Rennes, France
[2] MaIAGE, INRA, Université Paris-Saclay, Jouy-en-Josas, France
[3] CHU Pontchaillou, Rennes, France

## ABSTRACT

**Background**. Large scale metagenomic projects aim to extract biodiversity knowledge between different environmental conditions. Current methods for comparing microbial communities face important limitations. Those based on taxonomical or functional assignation rely on a small subset of the sequences that can be associated to known organisms. On the other hand, *de novo* methods, that compare the whole sets of sequences, either do not scale up on ambitious metagenomic projects or do not provide precise and exhaustive results.

**Methods**. These limitations motivated the development of a new *de novo* metagenomic comparative method, called Simka. This method computes a large collection of standard ecological distances by replacing species counts by *k*-mer counts. Simka scales-up today's metagenomic projects thanks to a new parallel *k*-mer counting strategy on multiple datasets.

**Results**. Experiments on public Human Microbiome Project datasets demonstrate that Simka captures the essential underlying biological structure. Simka was able to compute in a few hours both qualitative and quantitative ecological distances on hundreds of metagenomic samples (690 samples, 32 billions of reads). We also demonstrate that analyzing metagenomes at the *k*-mer level is highly correlated with extremely precise *de novo* comparison techniques which rely on all-versus-all sequences alignment strategy or which are based on taxonomic profiling.

Corresponding author
Gaëtan Benoit, gaetan.benoit@inria.fr

## INTRODUCTION

It is estimated that only a fraction of $10^{-24}$–$10^{-22}$ of the total DNA on earth has been sequenced (*Anonymous, 2011*). In large scale metagenomics studies such as *Tara* Oceans (*Karsenti et al., 2011*) most of the sequenced data comes from unknown organisms and their short reads assembly remains an inaccessible task (see for instance results from the CAMI challenge: http://cami-challenge.org/). When precise taxonomic assignation is not feasible, microbial ecosystems can nevertheless be compared on the basis of their diversity, inferred from metagenomic read sets. In this framework, the beta-diversity, introduced in *Whittaker (1960)*, measures the dissimilarities between communities in terms of species

composition. Such compositions may be approximated by sequencing marker genes, such as the rRNA 16S in bacterial communities (*Liles et al., 2003*), and clustering the sequences into Operational Taxonomic Units (OTU) or working species. However, marker genes surveys suffer from amplification and primer bias (*Cai et al., 2013*) and therefore may not capture the whole microbial diversity of a sample. Furthermore, even within the captured diversity, the marker may not be informative enough to discriminate between sub-species or even species strains (*Piganeau et al., 2011*). Finally, this approach is impractical for whole metagenomic sets for at least two reasons: clustering reads into putative species is computationally costly and leaves out a large fraction of the reads (*Nielsen et al., 2014*).

In this context, it is more practical to ditch species composition altogether and compare microbial communities using directly the sequence content of metagenomic read sets. This has first been performed by using Blast (*Altschul et al., 1990*) for comparing read content (*Yooseph et al., 2007*). This approach was successful but cannot scale up to large studies made up of dozens or hundreds of large read sets, such as those generated from Illumina sequencers.

In 2012, the Compareads method (*Maillet et al., 2012*) was proposed. The method compares the whole sequence content of two read sets. It introduced a rough approximation of read similarity based on the number of shared words of length $k$ ($k$-mer, with $k$ typically around 30) and used it for providing so defined similar reads between read sets. The number of similar reads was then used for computing a Jaccard distance between pairs of read sets. Commet (*Maillet et al., 2014*) is an extended version of Compareads. It better handles the comparison of large read sets and provides a read sub-set representation that facilitates result analyses and reduces the disk footprint. *Seth et al. (2014)* used the notion of shared $k$-mers between samples for estimating dataset similarities. This is a slightly different problem as this was used for retrieving from an indexed database, samples similar to a query sample. More recently, two additional methods were developed to represent a metagenome by a feature vector that is then used to compute pairwise similarity matrices between multiple samples. For both methods, features are based on the $k$-mer composition of samples, but with a feature representing more than one $k$-mer and using only a subset of $k$-mers to reduce the dimension (*Ulyantsev et al., 2016*; *Ondov et al., 2016*). However, the approaches for $k$-mer grouping and sub-sampling are radically different. In MetaFast (*Ulyantsev et al., 2016*), the subset of $k$-mers is obtained by post-processing *de novo* assemblies performed for each metagenome. A feature represents then a set of $k$-mers belonging to a same assembly graph "*component.*" The relative abundance of such component in each sample is then used to compute the Bray–Curtis dissimilarity measure. In Mash (*Ondov et al., 2016*), a sub-sampling of the $k$-mers is performed using the MinHash (*Broder, 1997*) approach (keeping by default 1,000 $k$-mers per sample). The method outputs then a Jaccard index of the presence-absence of such $k$-mers in two samples.

All these reference-free methods share the use of $k$-mers as the fundamental unit used for comparing samples. Actually, $k$-mers are a natural unit for comparing communities: (1) sufficiently long $k$-mers are usually specific of a genome (*Fofanov et al., 2004*); (2) $k$-mer frequency is linearly related to genome's abundance (*Wu & Ye, 2011*); (3) $k$-mer aggregates organisms with very similar $k$-mer composition (e.g., related strains from the

same bacterial species) without need for a classification of those organisms (*Teeling et al., 2004*). *Dubinkina et al. (2016)* conducted an extensive comparison between $k$-mer-based distances and taxonomic ones (i.e., based on taxonomic assignation against a reference database) for several large scale metagenomic projects. They demonstrate that $k$-mer-based distances are well correlated to taxonomic ones, and are therefore accurate enough to recover known biological structure, but also to uncover previously unknown biological features that were missed by reference-based approaches due to incompleteness of reference databases. Importantly, the greater $k$, the more correlated these taxonomic and $k$-mer-based distances seem to be. However, the study is limited to values of $k$ lower than 11 for computational reasons and the correlation for large values of $k$ still needs to be evaluated.

Even if Commet and MetaFast approaches were designed to scale-up to large metagenomic read sets, their use on data generated by large scale projects is turning into a bottleneck in terms of time and/or memory requirements. By contrast, Mash outperforms by far all other methods in terms of computational resource usage. However, this frugality comes at the expense of result quality and precision: the output distances and Jaccard indexes do not take into account relative abundance information and are not computed exactly due to $k$-mer sub-sampling.

In this paper, we present Simka. Simka compares $N$ metagenomic datasets based on their $k$-mers counts. It computes a large collection of distances classically used in ecology to compare communities. Computation is performed by replacing species counts by $k$-mer counts, for a large range of kmer sizes, including large ones (up to 30). Simka is, to our knowledge, the first method able to rapidly compute a full range of distances enabling the comparison of any number of datasets. This is performed by processing data on-the-fly (i.e., without storage of large temporary results). With the exception of Mash that is, thanks to sub-sampling, approximately two to five time faster, Simka outperforms state-of-the-art read comparison methods in terms of computational needs. For instance, Simka ran on 690 samples from the Human Microbiome Project (HMP) (*Human Microbiome Project Consortium, 2012a*) (totalling 32 billion reads) in less than 10 h and using no more than 70 GB RAM.

The contributions of this manuscript are three-fold. First we propose a new method for efficiently counting $k$-mers from a large number of metagenomic samples. The usefulness of such counting is not limited to comparative metagenomics and may have applications in many other fields. Second, we show how to derive a large number of ecological distances from $k$-mer counts. And third, we show on real datasets that $k$-mer-based distances are highly correlated to taxonomic distances: they therefore capture the same underlying structure and lead to the same conclusions.

## MATERIALS AND METHODS

The proposed algorithm enables to compute *dissimilarity measures* between read sets. In the following, in order to simplify the reading, we use the term "distance" to refer to this measure.

## Overview

Given $N$ metagenomic datasets, denoted as $S_1, S_2, S_i, \ldots, S_N$, the objective is to provide a $N \times N$ distance matrix $D$ where $D_{i,j}$ represents an ecological distance between datasets $S_i$ and $S_j$. Such possible distances are listed in Table 1. The computation of the distance matrix can be theoretically decomposed into two distinct steps:

1. $k$-**mer count**. Each dataset is represented as a set of discriminant features, in our case, $k$-mer counts. More precisely, a $k$-mer count matrix $KC$ of size $W \times N$ is computed. $W$ is the number of distinct $k$-mer among all the datasets. $KC_{i,j}$ represents the number of times a $k$-mer $i$ is present in the dataset $S_j$.

2. **distance computation**. Based on the $k$-mer count information, the distance matrix $D$ is computed. Actually, many ecological distances (cf Table 1) can be derived from matrix $KC$ when replacing species counts by $k$-mer counts.

Actually, Simka does not require to have the full $KC$ matrix to start the distance computation. However, for sake of simplicity, we will first consider this matrix to be available.

The $k$-mer count step splits all the reads of the datasets into $k$-mers and performs a global count. This can be done by counting individually $k$-mers in each dataset, then merging the overall $k$-mer counts. The output is the matrix $KC$ (of size $W \times N$). Efficient algorithms, such as KMC2 (*Deorowicz et al., 2015*), have recently been developed to count all the occurrences of distinct $k$-mers in a read dataset, allowing the computation to be executed in a reasonable amount of time and memory even on very large datasets. However, the main drawback of this approach is the huge main memory space it requires which is computed as follow: $Mem_{KC} = W_s * (8 + 4N)$ bytes, with $W_s$ the number of distinct $k$-mers, $N$ the number of samples, and 8 and 4 the number of bytes required to store respectively 31-mers and a $k$-mer count. For example, experiments on the HMP (*Human Microbiome Project Consortium, 2012a*) datasets (690 datasets containing on average 45 millions of reads each) would require a storage space of $260TB$ for the matrix $KC$.

However, a careful look at the definition of ecological distances (Table 1) shows that, up to some final transformation, they are all additive over the $k$-mers. Independent contributions to the distance can thus be computed in parallel from disjoint sets of $k$-mers and aggregated later on to construct the final distance matrix. Furthermore, each independent contribution can itself be constructed in an iterative way by receiving lines of the $KC$ matrix, called abundance vectors, one at a time. The abundance vector of a specific $k$-mer simply consists of its $N$ counts in the $N$ datasets.

To sum up, instead of computing the complete $k$-mer count matrix $KC$, the alternative computation scheme we propose is to generate successive abundance vectors from which independent contributions to the distances can be iteratively updated in parallel. The great advantage is that the huge $k$-mer count matrix $KC$ does not need to be stored anymore. However, this approach requires a new strategy to generate abundance vectors. We propose and describe below a new efficient multiset $k$-mer counting algorithm (called MKC) that can be highly parallelized on large computing resources infrastructures. As illustrated in Fig. 1, Simka uses abundance vectors generated by MKC for computing ecological distances.

**Table 1** **Definition of some classical ecological distances computed by Simka.** All quantitative distances can be expressed in terms of $C_S$, $f = f(x,y,X,Y)$ and $g = g(x)$, using the notations of Eq. (2), and computed in one pass. Qualitative ecological distances (resp. AB-variants of qualitative distances) can also be computed in a single pass over the data by computing first $a$, $b$ and $c$ (resp. $U$ and $V$). See main text for the definition of $a$, $b$, $c$, $U$ and $V$.

| Name | Definition | $C_{S_i}$ | $f(x,y,X,Y)$ | $g(x)$ |
|---|---|---|---|---|
| **Quantitative distances** | | | | |
| Chord | $\sqrt{2 - 2\sum_w \frac{N_{S_i}(w)N_{S_j}(w)}{C_{S_i}C_{S_j}}}$ | $\sqrt{\sum_w N_{S_i}(w)^2}$ | $\frac{xy}{XY}$ | $\sqrt{2-2x}$ |
| Hellinger | $\sqrt{2 - 2\sum_w \frac{\sqrt{N_{S_i}(w)N_{S_j}(w)}}{\sqrt{C_{S_i}C_{S_j}}}}$ | $\sum_w N_{S_i}(w)$ | $\frac{\sqrt{xy}}{\sqrt{XY}}$ | $\sqrt{2-2x}$ |
| Whittaker | $\frac{1}{2}\sum_w \frac{\left\lvert N_{S_i}(w)C_{S_j} - N_{S_j}(w)C_{S_i}\right\rvert}{C_{S_i}C_{S_j}}$ | $\sum_w N_{S_i}(w)$ | $\frac{\lvert xY-yX\rvert}{XY}$ | $\frac{x}{2}$ |
| Bray–Curtis | $1 - 2\sum_w \frac{\min(N_{S_i}(w),N_{S_j}(w))}{C_{S_i}+C_{S_j}}$ | $\sum_w N_{S_i}(w)$ | $\frac{\min(x,y)}{X+Y}$ | $1-2x$ |
| Kulczynski | $1 - \frac{1}{2}\sum_w \frac{(C_{S_i}+C_{S_j})\min(N_{S_i}(w),N_{S_j}(w))}{C_{S_i}C_{S_j}}$ | $\sum_w N_{S_i}(w)$ | $\frac{(X+Y)\min(x,y)}{XY}$ | $1-\frac{x}{2}$ |
| Jensen–Shannon | $\sqrt{\frac{1}{2}\sum_w \left[\frac{N_{S_i}(w)}{C_{S_i}}\log\frac{2C_{S_j}N_{S_i}(w)}{C_{S_j}N_{S_i}(w)+C_{S_i}N_{S_j}(w)} + \frac{N_{S_j}(w)}{C_{S_j}}\log\frac{2C_{S_i}N_{S_j}(w)}{C_{S_j}N_{S_i}(w)+C_{S_i}N_{S_j}(w)}\right]}$ | $\sum_w N_{S_i}(w)$ | $\frac{x}{X}\log\frac{2xY}{xY+yX} + \frac{y}{Y}\log\frac{2yX}{xY+yX}$ | $\sqrt{\frac{x}{2}}$ |
| Canberra | $\frac{1}{a+b+c}\sum_w \left\lvert\frac{N_{S_i}(w)-N_{S_j}(w)}{N_{S_i}(w)+N_{S_j}(w)}\right\rvert$ | – | $\left\lvert\frac{x-y}{x+y}\right\rvert$ | $\frac{1}{a+b+c}x$ |
| **Qualitative distances** | | | | |
| Chord/Hellinger | $\sqrt{2\left(1 - \frac{a}{\sqrt{(a+b)(a+c)}}\right)}$ | – | – | – |
| Whittaker | $\frac{1}{2}\left(\frac{b}{a+b} + \frac{c}{a+c} + \left\lvert\frac{a}{a+b} - \frac{a}{a+c}\right\rvert\right)$ | – | – | – |
| Bray–Curtis/Sorensen | $\frac{b+c}{2a+b+c}$ | – | – | – |
| Kulczynski | $1 - \frac{1}{2}\left(\frac{a}{a+b} + \frac{a}{a+c}\right)$ | – | – | – |
| Ochiai | $1 - \frac{a}{\sqrt{(a+b)(a+c)}}$ | – | – | – |
| Jaccard | $\frac{b+c}{a+b+c}$ | – | – | – |
| **Abundance-based (AB) variants of qualitative distances** | | | | |
| AB-Jaccard | $1 - \frac{UV}{U+V-UV}$ | – | – | – |
| AB-Ochiai | $1 - \sqrt{UV}$ | – | – | – |
| AB-Sorensen | $1 - \frac{2UV}{U+V}$ | – | – | – |

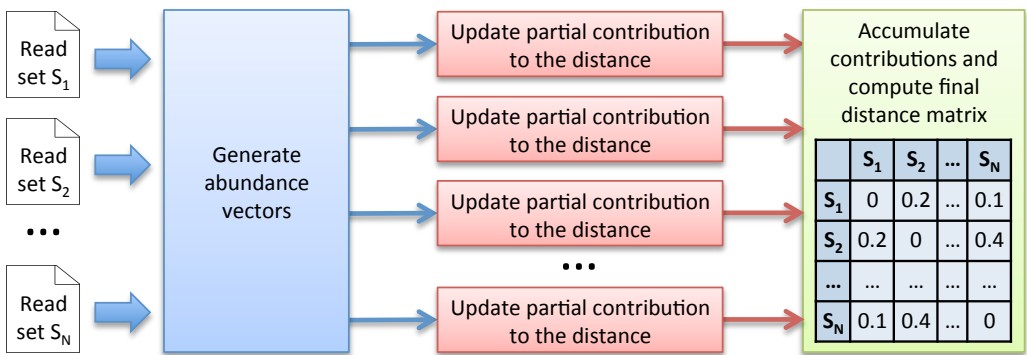

**Figure 1** **Simka strategy.** The first step takes as input $N$ datasets and generates multiple streams of abundance vector from disjoint sets of $k$-mers. The abundance vector of a $k$-mer consists of its $N$ counts in the $N$ datasets. These abundance vectors are taken as input by the second step to iteratively update independent contributions to the ecological distance in parallel. Once an abundance vector has been processed, there is no need to keep it on record. The final step aggregates each contribution and computes the final distance matrix.

## Multiset *k*-mer Counting

Starting from N datasets of reads, the aim is to generate abundance vectors that will feed the ecological distance computation step. This task is divided into two phases:

1. Sorting Count,
2. Merging Count.

*Sorting Count.* Each $k$-mer of a dataset is extracted and its canonical representation is stored (the canonical representation of a $k$-mer is the smallest lexicographic value between the $k$-mer and its reverse complement). Canonical $k$-mers are then sorted in lexicographical order. Distinct $k$-mers can thus be identified and their number of occurrences computed.

As the number of distinct $k$-mers is generally huge, the sorting step is divided into two sub-tasks and proceeds as follows: the $k$-mers are first separated into $P$ partitions, each stored on disk. After this preliminary task, each partition is sorted and counted independently, and stored again on disk. Conceptually, at the end of the sorting count process, we dispose of $N \times P$ sorted partitions. As the same distribution function is applied to all datasets, a partition $P_i$ contains a specific subset of $k$-mers common to all datasets. Figure 2A illustrates the Sorting Count phase.

The Sorting Count phase has a high parallelism potential. A first parallelism level is given by the independent counts of each dataset. $N$ processes can thus be run in parallel, each one dealing with a specific dataset. A second level is given by the fine grained parallelism implemented in software such as DSK (*Rizk, Lavenier & Chikhi, 2013*) or KMC2 (*Deorowicz et al., 2015*) that intensively exploit today multicore processor capabilities. Thus, the overall Sorting Count process is especially suited for grid infrastructures made of hundred of nodes, and where each node implements 8 or 16-core systems.

Furthermore, to limit disk bandwidth and avoid I/O bottleneck, partitions are compressed. A dictionary-based approach, such as the one provided in zlib (*Deutsch & Gailly, 1996*), is used. This type of compression is very well suited here since it efficiently packs the high redundancy of sorted $k$-mers.

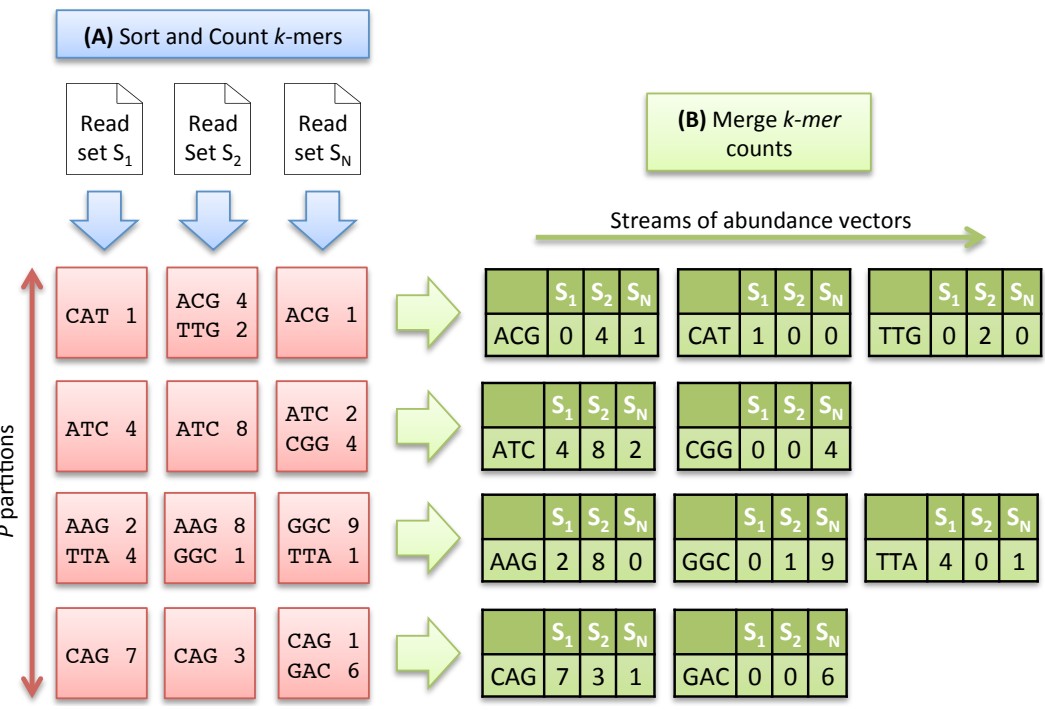

**Figure 2** **Multiset *k*-mer Counting strategy with *k* = 3.** (A) The sorting counting process, represented by a blue arrow, counts datasets independently. Each process outputs a column of *P* partitions (red squares) containing sorted *k*-mer counts. (B) The merging count process, represented by a green arrow, merges a row of *N* partitions. It outputs abundance vectors, represented in green, to feed the ecological distance computation process.

*Merging Count.* Here, the data partitioning introduced in the previous step is advantageously used to generate abundance vectors. The $N$ files associated to a partition $P_i$, are taken as input of a merging process. These files contain $k$-mer counts sorted in lexicographical order. A Merge-Sort algorithm can thus be efficiently applied to directly generate abundance vectors.

In that scheme, $P$ processes can be run independently, resulting in the generation of $P$ abundance vectors in parallel, allowing to compute simultaneously $P$ contributions of the ecological distance. Note that the abundance vectors do not need to be stored. They are only used as input streams for the next step. Figure 2B illustrates the Merging Count phase.

*k-mer abundance filter.* Distinct $k$-mers with very low abundance usually come from sequencing errors. As a matter of fact, a single sequencing error creates up to $k$ erroneous distinct $k$-mers. Filtering out these $k$-mers speeds-up the Simka process, as it greatly reduces the overall number of distinct $k$-mers, but may also impact the content of the distance matrix. This point is evaluated and discussed in the result section.

This filter is activated during the count process. Only $k$-mers whose abundance is equal to or greater than a given abundance threshold are kept. By default the threshold is set to 2. The $k$-mers that pass the filter are called "solid $k$-mers."

## Ecological distance computation

Simka computes a collection of distances for all pairs of datasets. As detailed in the previous section, abundance vectors are used as input data. For the sake of simplicity, we first explain the computations of the Bray–Curtis distance. All other distances, presented later on, can be computed in the same way, with only small adaptations.

*Computing the Bray–Curtis distance.* The Bray–Curtis distance is given by the following equation:

$$BrayCurtis_{Ab}(S_i, S_j) = 1 - 2\frac{\sum_{w \in S_i \cap S_j} min(N_{S_i}(w), N_{S_j}(w))}{\sum_{w \in S_i} N_{S_i}(w) + \sum_{w \in S_j} N_{S_j}(w)} \tag{1}$$

where $w$ is a $k$-mer and $N_{S_i}(w)$ is the abundance of $w$ in the dataset $S_i$. We consider here that $w \in S_i \cap S_j$ if $N_{S_i}(w) > 0$ and $N_{S_j}(w) > 0$.

The equation involves marginal (or dataset specific) terms (i.e., $\sum_{w \in S_i} N_{S_i}(w)$ is the total amount of $k$-mers in dataset $S_i$) acting as normalizing constants and crossed terms that capture the (dis)similarity between datasets (i.e., $\sum_{w \in S_i \cap S_j} min(N_{S_i}(w), N_{S_j}(w))$ is the total amount of $k$-mers in the intersection of the datasets $S_i$ and $S_j$). Marginal and crossed terms are then combined to compute the final distance.

Algorithm 1 shows that it is straightforward to compute the distance matrix between $N$ datasets from the abundance vectors. Inputs of this algorithm are provided by the Multiple $k$-mer Counting algorithm (MKC). These are the $P$ streams of abundance vectors and the marginal terms of the distance, i.e., the number of $k$-mers in each dataset, determined during the first step of the MKC which counts the $k$-mers.

A matrix, denoted $M_\cap$, of dimension $N \times N$ is initialized (step 1) to record the final value of the crossed terms of each pair of datasets. $P$ independent processes are run (step 2) to compute $P$ partial crossed term matrices, denoted $M_{\cap part}$ (step 3), in parallel. Each process iterates over its abundance vector stream (step 4). For each abundance vector, we loop over each possible pair of datasets (steps 5–6). The matrix $M_{\cap part}$ is updated (step 8) if the $k$-mer is shared, meaning that it has positive abundance in both datasets $S_i$ and $S_j$ (step 7). Since a distance matrix is symmetric with null diagonal, we limit the computation to the upper triangular part of the matrix $M_{\cap part}$. The current abundance vector is then released. Each process writes its matrix $M_{\cap part}$ on the disk when its stream is done (step 9).

When all streams are done, the algorithm reads each written $M_{\cap part}$ and accumulates it to $M_\cap$ (step 10–11). The last loop (steps 13–16) computes the Bray–Curtis distance for each pair of datasets and fills the distance matrix reported by Simka.

The amount of abundance vectors streamed by the MKC is equal to $W_s$, which is also the total amount of distinct solid $k$-mers in the $N$ datasets. This algorithm has thus a time complexity of $O(W_s \times N^2)$.

*Other ecological distances.* The distance introduced in Eq. (1) is a single example of ecological distance. There exists numerous other ecological distances that can be broadly classified into two categories (see *Legendre & De Cáceres (2013)* for a finer classification): distances based on presence-absence data (hereafter called *qualitative*) and distances

---

**Algorithm 1:** Compute the Bray-Curtis distance (Eq. (1)) between $N$ datasets

**Input:**
- $V_s$: vector of size $P$ representing the abundance vector streams
- $V_\cup$: vector of size $N$ containing the number of $k$-mers in each dataset

**Output:** a distance matrix *Dist*

1  $M_\cap \leftarrow$ empty square matrix of size $N$ // number of $k$-mers in each dataset intersection

2  **In parallel: foreach** *abundance vector stream S* **in** $V_s$ **do**

3      $M_{\cap part} \leftarrow$ empty squared matrix of size $N$ // part of $M_\cap$

4      **foreach** *abundance vector v* **in** *S* **do**

5          **for** $i \leftarrow 0$ **to** $N-1$ **do**

6              **for** $j \leftarrow i+1$ **to** $N-1$ **do**

7                  **if** $v[i] > 0$ *and* $v[j] > 0$ **then**

8                      $M_{\cap part}[i,j] \leftarrow M_{\cap part}[i,j] + \min(v[i], v[j])$

9      Write $M_{\cap part}$ to disk

10  **foreach** *each* written matrix $M_{\cap part}$ **do**

11      $M_\cap \leftarrow M_\cap + M_{\cap part}$

12  *Dist* $\leftarrow$ empty squared matrix of size $N$ // final distance matrix

13  **for** $i \leftarrow 0$ **to** $N-1$ **do**

14      **for** $j \leftarrow i+1$ **to** $N-1$ **do**

15          $Dist[i,j] = 1 - 2 * M_\cap[i,j] / (V_\cup[i] + V_\cup[j])$

16          $Dist[j,i] = 1 - 2 * M_\cap[i,j] / (V_\cup[i] + V_\cup[j])$

17  **return** *Dist*

---

based on proper abundance data (hereafter called *quantitative*). Qualitative distances are more sensitive to factors that affect presence-absence of organisms (such as pH, salinity, depth, humidity, absence of light, etc.) and therefore useful to study bioregions. Quantitative distances focus on factors that affect relative changes (seasonal changes, nutrient availability, concentration of oxygen, depth, diet, disease, etc.) and are therefore useful to monitor communities over time or along an environmental gradient. Note that some factors, such as pH, are likely to affect both presence-absence (for large changes in pH) and relative abundances (for small changes in pH). Algorithmically, most ecological distances, including most of those mentioned in *Legendre & De Cáceres (2013)*, can be expressed for two datasets $S_i$ and $S_j$ as:

$$Distance(S_i, S_j) = g\left(\sum_{w \in S_i \cup S_j} f\left(N_{S_i}(w), N_{S_j}(w), C_{S_i}, C_{S_j}\right)\right) \tag{2}$$

where $g$ and $f$ are simple functions, and $C_{S_i}$ is a marginal (i.e., dataset-specific) term of dataset $S_i$, usually of size 1 (i.e., a scalar). In most distances, $C_{S_i}$ is simply the total number

of $k$-mers in $S_i$. By contrast, the value of $f$ corresponds to crossed terms and requires knowledge of both $N_{S_i}(w)$ and $N_{S_j}(w)$ (and potentially $C_{S_i}$ and $C_{S_j}$ as well). For instance, for the abundance-based Bray–Curtis distance of Eq. (1), we have $C_{S_i} = \sum_{w \in S_i} N_{S_i}(w)$, $g(x) = 1 - 2x$ and $f(x, y, X, Y) = \min(x, y)/(X + Y)$. Those distances can be computed in a single pass over the data using a slightly modified variant of Algorithm 1. The marginal terms $C_{S_i}$ are computed during the first step of the MKC which counts the $k$-mers of each dataset. The crossed terms involving $f$ are computed and summed in steps 7–8 (but exact instructions depend on the nature of $f$). Finally, the actual distances are computed in steps 15–16 and depend on both $f$ and $g$.

Qualitative distances form a special case of ecological distances: they can all be expressed in terms of quantities $a$, $b$ and $c$ where $a$ is the number of distinct $k$-mers shared between datasets $S_i$ and $S_j$, $b$ is the number of distinct $k$-mers specific to dataset $S_i$ and $c$ is the number of distinct $k$-mers specific to dataset $S_j$. Those distances easily fit in the previous framework as $a = \sum_{w \in S_i \cap S_j} 1_{\{N_{S_i}(w) N_{S_j}(w) > 0\}}$, $C_{S_i} = \sum_{w \in S_i} 1_{\{N_{S_i}(w) > 0\}} = a + b$ and similarly $C_{S_j} = a + c$. Therefore, $a$ is a crossed term and $b$ and $c$ can be deduced from $a$ and the marginal terms.

In the same vein, *Chao et al. (2006)* introduced variations of presence-absence distances incorporating abundance information to account for unobserved species. The main idea is to replace "hard" quantities such as $a/(a+b)$, the fraction of distinct $k$-mers from $S_i$ shared with $S_j$, by probabilistic "soft" ones: here the probability $U \in [0, 1]$ that a $k$-mer from $S_i$ is also found in $S_j$. Similarly, the "hard" fraction $a/(a+c)$ of distinct $k$-mers from $S_j$ shared with $S_i$ is replaced by the "soft" probability $V$ that a $k$-mer from $S_j$ is also found in $S_i$. $U$ and $V$ play the same role as $a$, $b$ and $c$ do in qualitative distances and are sufficient to compute the variants named AB-Jaccard, AB-Ochiai and AB-Sorensen. However and unlike the quantities $a$, $b$ $c$, which can be observed from the data, $U$ and $V$ are not known in practice and must be estimated from the data. *Chao et al. (2006)* proposed several estimates for $U$ and $V$. The most elaborate ones attempt to correct for differences in sampling depths and unobserved species by considering the complete $k$-mer counts vector of a sample. Those estimates are unfortunately untractable in our case as we stream only a few $k$-mer counts at a time. Instead we resort to the simplest estimates presented in *Chao et al. (2006)*, which lend themselves well to the additive and distributed nature of Simka: $U = Y_{S_i S_j}/C_{S_i}$ and $V = Y_{S_j S_i}/C_{S_j}$ where $Y_{S_i S_j} = \sum_{w \in S_i \cap S_j} N_{S_i}(w) 1_{\{N_{S_j}(w) > 0\}}$ and $C_{S_i} = \sum_{w \in S_i} N_{S_i}(w)$. Note that $Y_{S_i S_j}$ corresponds to crossed terms and is asymmetric, i.e., $Y_{S_i S_j} \neq Y_{S_j S_i}$. Intuitively, $U$ is the fraction of $k$-mers (not distinct anymore) from $S_i$ also found in $S_j$ and therefore gives more weights to abundant $k$-mers that its qualitative counterpart $a/(a+b)$.

Table 1 gives the definitions of the collection of distances computed by Simka while replacing species counts by $k$-mer counts. These are qualitative, quantitative and abundance-based variants of qualitative ecological distances. The table also provides their expression in terms of $C_i$, $f$ and $g$, adopting the notations of Eq. (2).

Finally, note that the additive nature of the computed distances over $k$-mers is instrumental in achieving a linear time complexity (in $W_s$, the amount of distinct solid $k$-mers) and in the parallel nature of the algorithm. The algorithm is therefore not amenable to other, more complex classes of distances that account for ecological similarities between

species (*Pavoine et al., 2011*), or edit distances between $k$-mers as those complex distances require all versus all $k$-mer comparisons.

## Implementation

Simka is based on the GATB library (*Drezen et al., 2014*), a C++ library optimized to handle very large sets of $k$-mers. It includes a powerful implementation of the sorting count algorithm with the latest improvements in terms of $k$-mer counting introduced by *Deorowicz et al. (2015)*.

Simka is usable on standard computers and has also been entirely parallelized for grid or cloud systems. It automatically splits the process into jobs according to the available number of nodes and cores. These jobs are sent to the job scheduling system, while the overall synchronization is performed at the data level.

Simka is an open source software, distributed under GNU affero GPL License, available for download at https://gatb.inria.fr/software/simka/.

## RESULTS

First, Simka performances are evaluated in terms of computation time, memory footprint and disk usage and compared to those of other state of the art methods. Then, the Simka distances are evaluated with respect to *de novo* and reference-based distances and with respect to known biological results.

We conduct our numerical experiments on data from the Human Microbiome Project (HMP) (*Human Microbiome Project Consortium, 2012a*) which is currently one of the largest publicly available metagenomic datasets: 690 samples gathered from different human body sites (http://www.hmpdacc.org/HMASM/). The whole dataset contains 2*16 billions of Illumina paired reads distributed non uniformly across the 690 samples. One advantage of this dataset is that it has been extensively studied, in particular the microbial communities are relatively well represented in reference databases (*Human Microbiome Project Consortium, 2012a*; *Human Microbiome Project Consortium, 2012b*) (see http://hmpdacc.org/pubs/publications.php for a complete list). Article S1 details precisely how the datasets used for each experiment were built.

### Performance evaluation

*Performances on small datasets.* The scalability of Simka was first evaluated on small subsets of the HMP project, where the number of compared samples varied from 2 to 40. When computing a simple distance, such as Bray–Curtis for instance, Simka running time shows a linear behavior with the number of compared samples (Fig. 3A). As expected, counting the kmers for each sample (MKC-count) consumes most of the time. This task has a theoretical time complexity linear with the number of kmers, and thus the number of samples, and this explains the observed linear behavior of the overall program. In fact, most steps of Simka, namely MKC-count, MKC-merge and simple distance computation, show a linear behavior between running time and the number of compared samples. The only exception is the computation of complex distances, where the time devoted to this task increases quadratically. Both simple and complex distance computation algorithms have

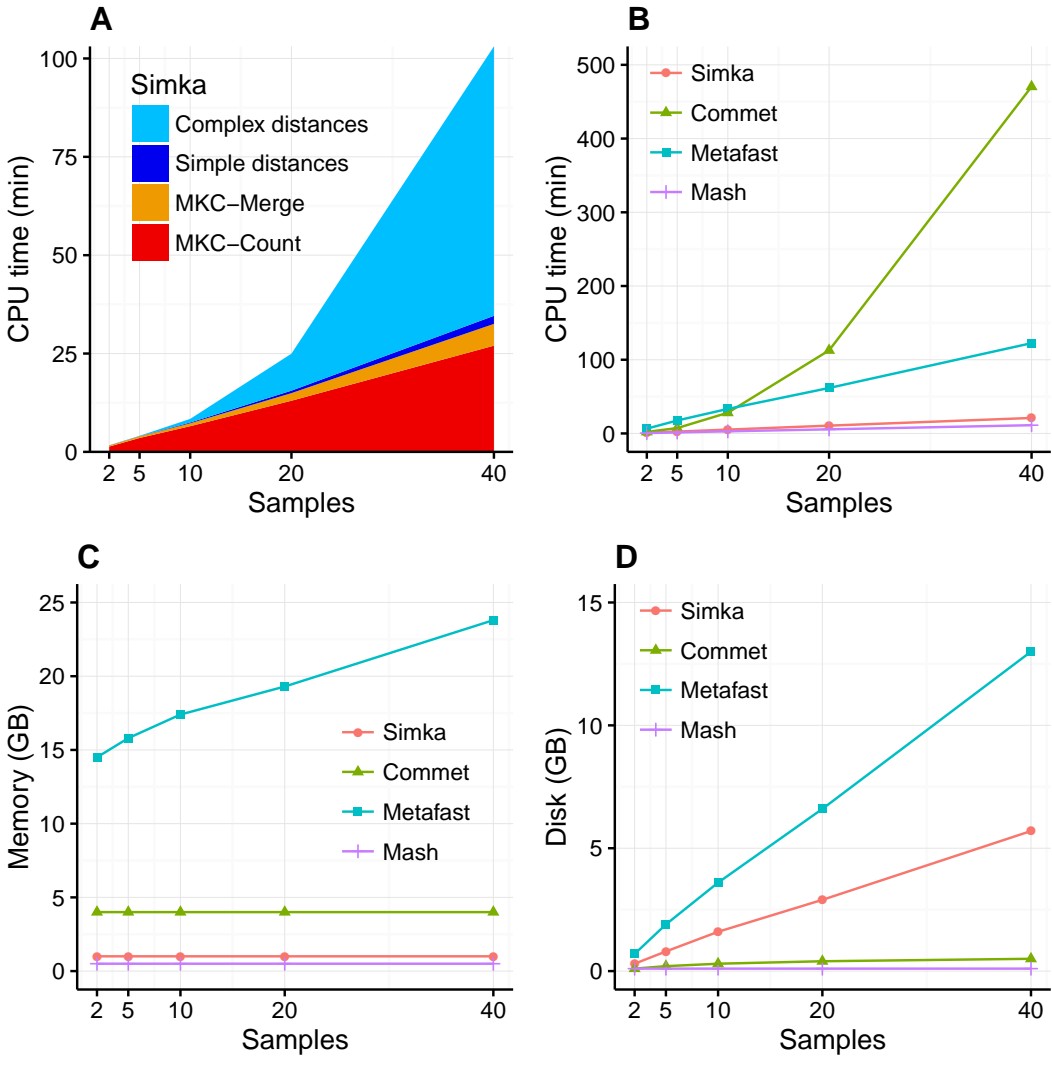

**Figure 3** **Simka performances with respect to the number N of input samples.** Each dataset is composed of two million reads. All tools were run on a machine equipped with a 2.50 GHz Intel E5-2640 CPU with 20 cores, 264 GB of memory. (A) and (B) CPU time with respect to $N$. For (A), colors correspond to different main Simka steps. (C) Memory footprint with respect to $N$. (D) Disk usage with respect to $N$. Parameters and command lines used for each tool are detailed in Table S1.

theoretical worst case quadratic time complexity relatively to $N$ (the number of samples). The difference of execution time comes then from the amount of operations required, in practice, to calculate the crossed terms of the distances. For a given abundance vector, the simple distances only need to be updated for each pair $(S_i, S_j)$ such that $N_{S_i} > 0$ and $N_{S_j} > 0$ whereas complex distances need to be updated for each pair such that $N_{S_i} > 0$ or $N_{S_j} > 0$, entailing a lot more update operations. It is noteworthy that among all distances listed in Table 1, all distances are simple, except the Whittaker, Jensen–Shannon and Canberra distances.

When compared to other state of the art tools, namely Commet, Metafast and Mash, we parameterized Simka to compute only the Bray–Curtis distance, since all other tools

**Table 2 Simka performances and *k*-mer statistics of the whole HMP project (690 samples).** Simka was run on a machine equipped with a 2.50 GHz Intel E5-2640 CPU with 20 cores, 264 GB of memory, with $k = 31$. Numbers of distinct *k*-mers are computed before and after the MKC-Merge algorithm: the *before merging* number is obtained by summing over all samples the distinct *k*-mers computed for each sample independently, whereas in the *after merging* number, *k*-mers shared by several samples are counted only once. Line "*Total time*" does not include complex distances whose computation is optional.

| HMP-690 samples-3727 GB-2×16 billion paired reads | | |
|---|---|---|
| | **Without filter** | **With filter** |
| Number of *k*-mers | $2471 \times 10^9$ | $2331 \times 10^9$ |
| Number of distinct *k*-mers before merging | $251 \times 10^9$ | $111 \times 10^9$ |
| Number of distinct *k*-mers after merging | $95 \times 10^9$ | $15 \times 10^9$ |
| Memory (GB) | 62 | 62 |
| Disk (GB) | 1,661 | 795 |
| Total time (min) | 1,338 | 862 |
| MKC-Count (min) | 758 | 573 |
| MKC-Merge (min) | 148 | 77 |
| Simple distances (min) | 432 | 212 |
| Complex distances (min) | 8,957 | 4,160 |

compute only one such simple distance. The Figs. 3B–3D shows respectively the CPU time, the memory footprint and the disk usage of each tool with respect to an increasing number of samples $N$. Mash has definitely the best scalability but limitations of its computed distance are shown in the next section. Commet is the only one to show a quadratic time behaviour with $N$. For $N = 40$, Simka is 6 times faster than Metafast and 22 times faster than Commet. All tools, except Metafast, have a constant maximal memory footprint with respect to $N$. For metafast, we could not use its max memory usage option since it often created "out of memory" errors. The disk usage of the four tools increases linearly with $N$. The linear coefficient is greater for Simka and MetaFast, but it remains reasonable in the case of Simka, as it is close to half of the input data size, which was 11 GB for $N = 40$.

In summary, Simka and Mash seems to be the only tools able to deal with very large metagenomics datasets, such as the full HMP project.

*Performances on the full HMP samples.* Remarkably, on the full dataset of the HMP project (690 samples), the overall computation time of Simka is about 14 h with very low memory requirements (see Table 2). By comparison, Metafast ran out of memory (it also ran out of memory while considering only a sub-sample composed of the 138 HMP gut samples) and Commet took several days to compute one 1-vs-all distance matrix and therefore would require years of computation to achieve the $N \times N$ distance matrix. Conversely, Mash ran in less than 5 h (255 min) and is faster than Simka. This was expected since Mash outputs an approximation of a simple qualitative distance, based on a sub-sample of 10,000 *k*-mers. By comparison, Simka computes numerous distances, including quantitative ones, over 15 billion distinct *k*-mers (see Table 2). Note that Simka is also designed for coarse-grain parallelism, and such computation took less than 10 h on a 200-CPU platform.

These results were obtained with default parameters, namely filtering out $k$-mers seen only once. On this dataset, this filter removes only 5 % of the data: solid $k$-mers ($k$-mers seen at least twice) account for 95% of all base pairs of the whole dataset (see Table 2). But interestingly, when speaking in terms of distinct $k$-mers, solid distinct $k$-mers represent less than half of all distinct $k$-mers before merging across all samples and only 15% after merging. Consequently, Simka performances are greatly improved, both in terms of computation time and disk usage when considering only solid $k$-mers. Notably, this does not degrade distance quality, at least for the HMP dataset, as shown in the next section. Additional tests on the impact of $k$ on the performances show that the disk usage increases sub-linearly with $k$ whereas the computation time and the memory usage stay constant (see Fig. S1).

## Evaluation of the distances

We evaluate the quality of the distances computed by Simka answering two questions. First, are they similar to distances between read sets computed using other approaches? Second, do they recover the known biological structure of HMP samples? For the first evaluation, two types of other approaches are considered, either *de novo* ones (similar to Simka but based on read comparisons), or taxonomic distances, e.g. approaches based on a reference database.

*Correlation with read-based approaches.* In this section, we focus on comparing Simka $k$-mer-based distance to two read-based approaches: Commet (*Maillet et al., 2014*) and an alignment-based method using BLAT (*Kent, 2002*). Both these read-based approaches define and use a read *similarity* notion. They derive the percentage of reads from one sample *similar* to at least one read from the other sample as a quantitative similarity measure between samples. Commet considers that two reads are similar if they share at least $t$ non-overlapping $k$-mers (here $t = 2$, $k = 33$). For BLAT alignments, similarity was defined based on several identity thresholds: two reads were considered similar if their alignment spanned at least 70 nucleotides and had a percentage of identity higher than 92%, 95% or 98%. For ease of comparison, Simka distance was transformed to a similarity measure, such as the percentage of shared kmers (see Article S1 for details of transformation).

Looking at the correlation with Commet is interesting because this tool uses a heuristic based on shared $k$-mers but its final distance is expressed in terms of read counts. As shown in Fig. 4, on a dataset of 50 samples from the HMP project, Simka and Commet similarity measures are extremely well correlated (Spearman correlation coefficient $r = 0.989$).

Similarly, clear correlations ($r > 0.89$) are also observed between the percentage of matched $k$-mers and the percentage of similar reads as defined by BLAT alignments (Fig. 5). Interestingly, the correlation depends on the $k$-mer size and the identity threshold used for BLAT: larger $k$-mer sizes correlate better with higher identity thresholds and *vice versa*. The highest correlation is 0.987, obtained for Simka with $k = 21$ compared to BLAT results with 95% identity.

These results demonstrate that we can safely replace read-based metrics by a kmer-based one, and this enables to save huge amounts of time when working on large metagenomics

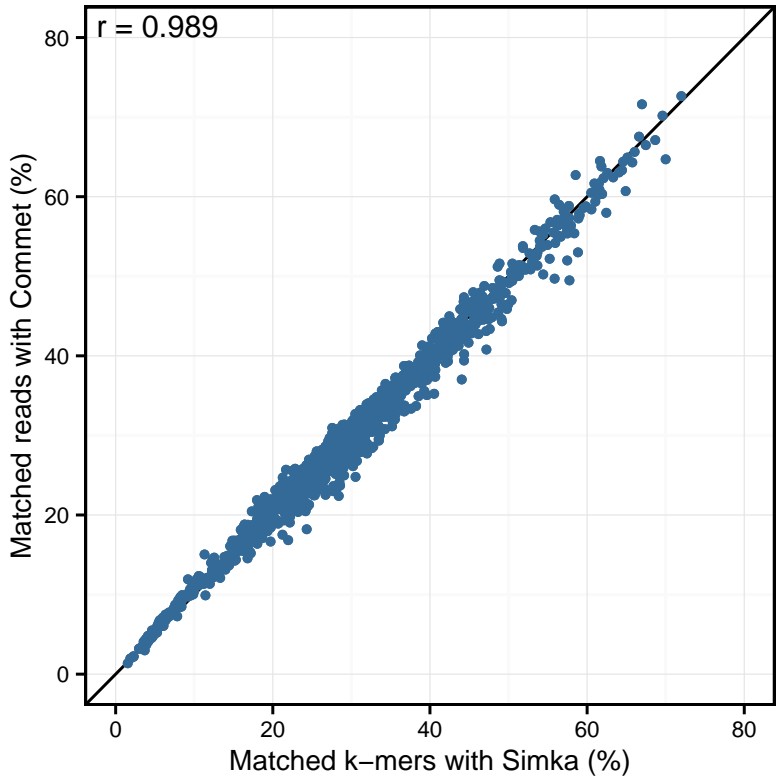

**Figure 4  Comparison of Simka and Commet similarity measures.** Commet and Simka were both used with Commet default $k$ value ($k = 33$). In this scatterplot, each point represents a pair of samples, whose $X$ coordinate is the % of matched $k$-mers computed by Simka, and the $Y$ coordinate is the % of matched reads computed by Commet.

projects. Moreover, the $k$-mer size parameter seems to be the counterpart of the identity threshold of alignment-based methods if one wants to tune the taxonomic precision level of the comparisons.

*Correlation with taxonomic distances on the gut sample.* A traditional way of comparing metagenomics samples rely on so called taxonomic distances that are based on sequence assignation to taxons by mapping to reference databases. To compare Simka to such traditional reference-based method, we used the HMP gut samples, which is a well studied dataset comprising 138 samples. The HMP consortium provides a quantitative taxonomic profile for each sample on its website. These profiles were obtained by mapping the reads on a reference genome catalog at 80% of identity. From these profiles, we computed the Bray–Curtis distance, latter used as a reference. The complete protocol to obtain taxonomic distances is given in Article S1. Only Mash and Simka results have been considered for this experiment. As previously mentioned, Commet and MetaFast could not scale this dataset.

Simka $k$-mer-based distance appears very well correlated to the traditional taxonomic distance ($r = 0.89$, see Fig. 6). On this figure, one may also notice that Simka measures are robust with the whole range of distances. On the other hand, Mash distances correlate badly with taxonomic ones ($r = 0.51$, see Fig. S3 and the comparison protocol in Article S1). This

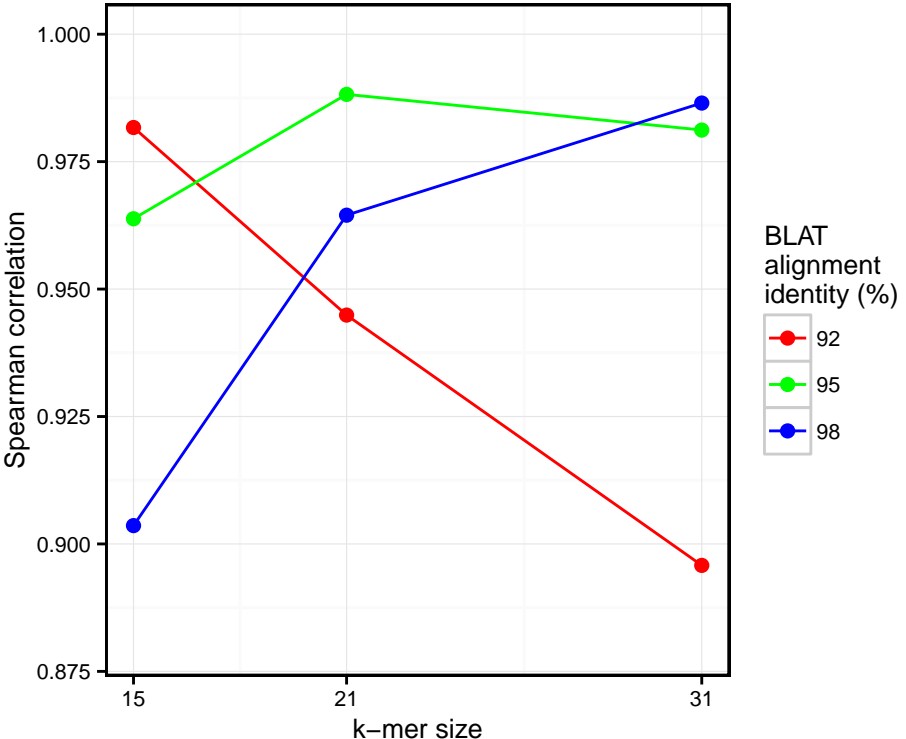

**Figure 5 Comparison of Simka and BLAT distances for several values of $k$ and several BLAT identity thresholds.** Spearman correlation values are represented with respect to $k$. The scatterplots obtained for each point of this figure are shown in Fig. S2.

is probably due to the fact that gut samples differ more in terms of relative abundances of microbes than in terms of composition (see next section). As Mash can only output a qualitative distance, it is ill equipped to deal with such a case. Additionally, as shown in Fig. S3, this conclusion stands for the HMP samples from other body sites for which one disposes of high quality taxonomic distances.

Interestingly, these Simka results are robust with the $k$-mer filtering option and the $k$-mer size, as long as $k$ is larger than 15 and with an optimal correlation obtained with $k = 21$ (see Fig. S4). Notably, with very low values of $k$ ($k < 15$), the correlation drops ($r = 0.5$ for $k = 12$). This completes previous results suggesting that the larger the $k$ the better the correlation, that were limited to $k$ values smaller than 11 (*Dubinkina et al., 2016*).

*Visualizing the structure of the HMP samples.* We propose to visualize the structure of the HMP samples and see if Simka is able to reproduce known biological results. To easily visualize those structures, we used the Principal Coordinate Analysis (PCoA) (*Borg & Groenen, 2013*) to get a 2-D representation of the distance matrix and of the samples: distances in the 2-D plane optimally preserve values of the distance matrix.

Figure 7 shows the PCoA of the quantitative Ochiai distance computed by Simka on the full HMP samples. We can see that the samples are clearly segregated by body sites. This is in line with results from studies of the HMP consortium (*Human Microbiome Project Consortium, 2012a*; *Costello et al., 2009*; *Koren et al., 2013*). Moreover, one may notice that

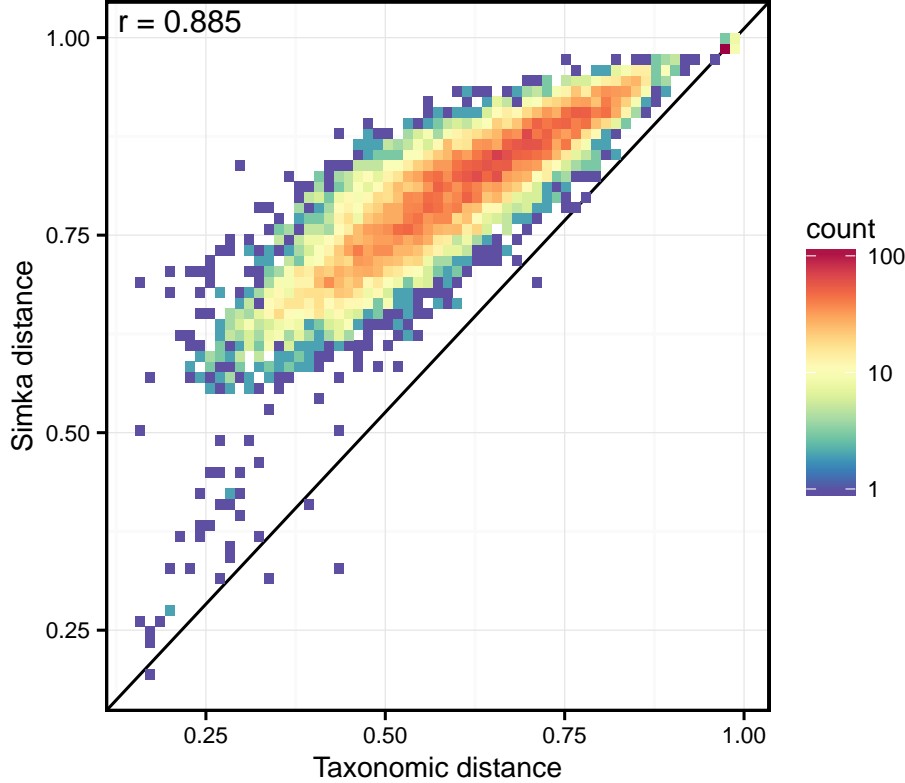

**Figure 6** **Correlation between taxonomic distance and *k*-mer based distance computed by Simka on HMP gut samples.** On this density plot, each point represents one or several pairs of the gut samples. The *X* coordinate indicates the Bray–Curtis taxonomic distance, and the *Y* coordinate the Bray–Curtis distance computed by Simka with $k = 21$. The color of a point is function of the amount of sample pairs with the given pair of distances (log-scaled).

different distances can lead to different distributions of the samples, with some clusters being more or less discriminated (see Fig. S5). This confirms the fact that it is important to conduct analyses using several distances as suggested in (*Koren et al., 2013*; *Legendre & De Cáceres, 2013*) as different distances may capture different features of the samples.

We conduct the same experiment on the 138 gut samples from the HMP project. *Arumugam et al. (2011)* showed that the gut samples are organized in three groups, known as enterotypes, and characterized by the abundance of a few genera: *Bacteroides*, *Prevotella* and genera from the *Ruminococcaceae* family. The original enterotypes were built from Jensen–Shannon distances on taxonomic profiles. The Fig. 8 shows the PCoA of the Jensen–Shannon distances obtained with Simka. Mapping the relative abundance of those genera in each sample, as provided by the HMP consortium, on the 2-D representation reveals a clear gradient in the PCoA space. Simka distances therefore recover biological features it had no direct access to: here, the fact that gut samples are structured along gradients of *Bacteroides*, *Prevotella* and *Ruminococcaceae*. The fact that Simka is able to capture such subtle signal raises hope of drawing new interesting biological insights from the data, in particular for those metagenomics project lacking good references (soil, seawater for instance).

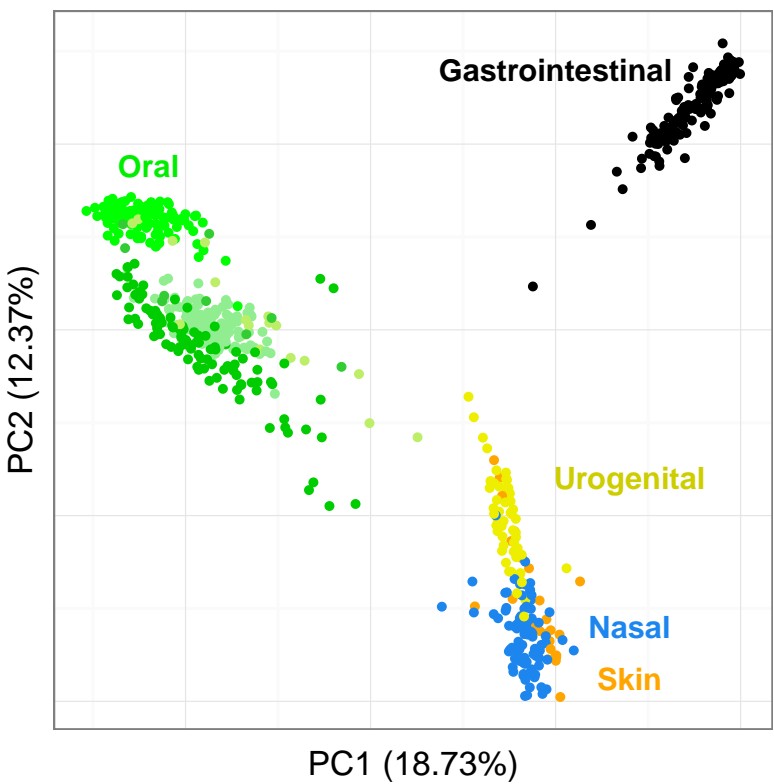

**Figure 7  Distribution of the diversity of the HMP samples by body sites.** PCoA of the samples is based on the quantitative Ochiai distance computed by Simka with $k = 21$. Each dot corresponds to a sample and is coloured according to the human body site it was extracted from. The green color shades correspond to three different subparts of the Oral samples: Tongue dorsum, Supragingival plaque, Buccal mucosa (from top to down).

## DISCUSSIONS

In this article, we introduced Simka, a new method for computing a collection of ecological distances, based on $k$-mer composition, between many large metagenomic datasets. This was made possible thanks to the Multiple $k$-mer Count algorithm (MKC), a new strategy that counts $k$-mers with state-of-the-art time, memory and disk performances. The novelty of this strategy is that it counts simultaneously $k$-mers from any number of datasets, and that it represents results as a stream of data, providing counts in each dataset, $k$-mer per $k$-mer.

The distance computation has a time complexity in $O(W \times N^2)$, with $W$ is the number of considered distinct $k$-mers and $N$ is the number of input samples. $N$ is usually limited to a few dozens or hundreds and cannot be reduced. However, $W$ may range in the hundreds of billions. The solid filter already provides large speed improvement without affecting the results, at least on the tests performed on the HMP datasets. However, the HMP dataset is not representative of all metagenomics projects and, in some cases, this filter may not be desired; for instance, in the case of samples with low coverage or when performing qualitative studies where the rare species have more impact. As a matter of fact,

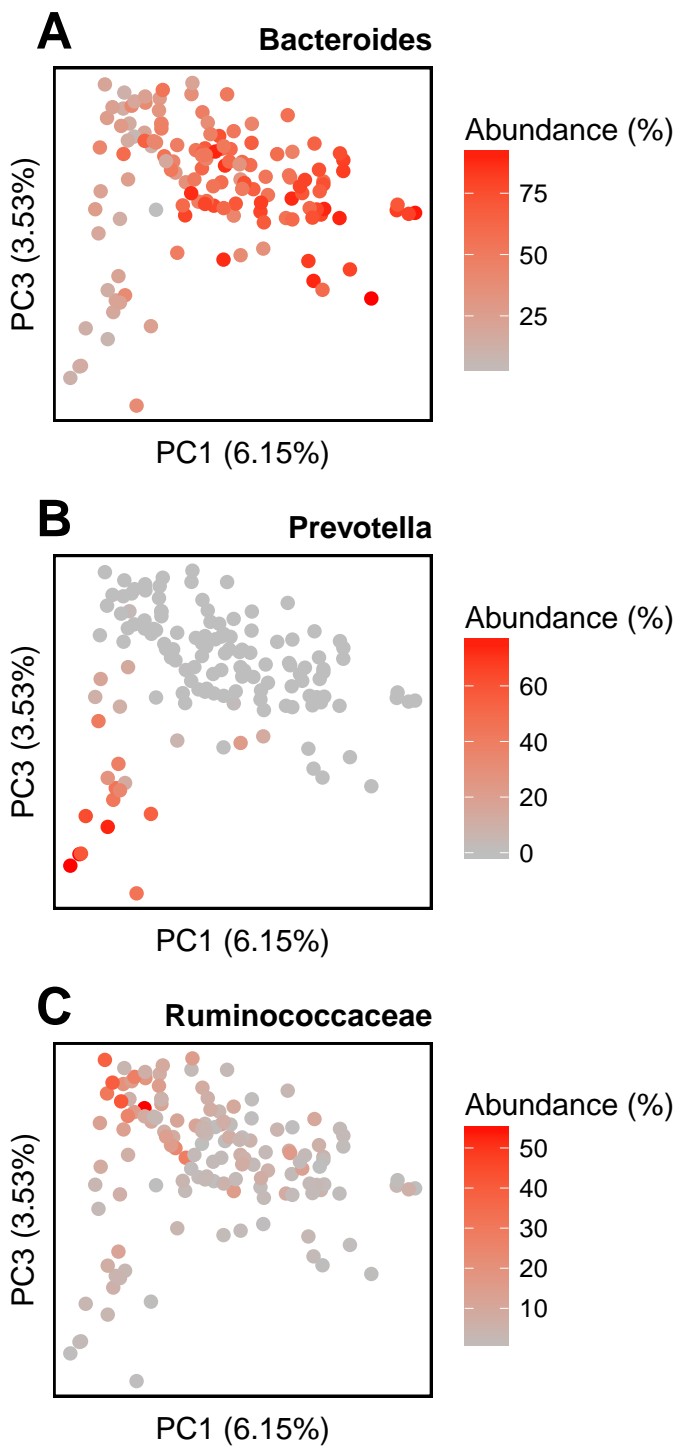

**Figure 8  Relative abundances of main genera in HMP gut samples.** Distribution of the gut samples from the HMP project is shown in a PCoA of the Jensen–Shannon distance matrix. This distance matrix was computed by Simka with $k = 21$. Relative abundances (0–100%) of (A) *Bacteroides*, (B) *Prevotella* and (C) *Ruminococcaceae*, as computed with Metaphlan (*Segata et al., 2012*), are mapped onto the sample points as color shades.

it is notable that Simka is able to scale large datasets even with the solid filter disabled as shown in the performance section. Interestingly, when applied on a low coverage dataset, namely the Global Ocean Sampling (*Yooseph et al., 2007*), Simka was able to capture the essential underlying biological structure with or without the $k$-mer solid filter (see Fig. S6). However, an important incoming challenge is to precisely measure the impact of applied thresholds together with the choice of $k$, depending on the input dataset features such as community complexity and sequencing effort.

Since metagenomic projects are constantly growing, it is important to offer the possibility to add new sample(s) to a set for which distances are already computed, without starting back the whole computation from scratch. It is straightforward to adapt the MKC algorithm to such operation, but the merging step and distance computation step have to be done again. However, adding a new sample does not modify previously computed distances and only requires to compute a single line of the distance matrix, it can thus be achieved in linear time.

The motivation for computing a collection of distances rather than just one is two folds: different distances capture different features of the data (*Koren et al., 2013*; *Legendre & De Cáceres, 2013*; *Pavoine et al., 2011*) and all the distances computed by Simka have in common that they are additive over $k$-mers and can thus be computed simultaneously using the same algorithm. To support the first point, we have seen that Mash performed badly when considering HMP samples per body sites since this tool can only take into account presence/absence information and not relative abundances in contrast to Simka. As a matter of fact, differences in relative abundances are subtler signals that are often at the heart of interesting biological insights in comparative genomics studies. For instance, *Boutin et al. (2015)* showed that the structure between different samples from lung disease patients was visible with the Bray Curtis (quantitative) distance and absent with the qualitative Jaccard distance, highlighting the role of the abundances of certain pathogenic microbes in the disease. In other studies, the response of bacterial communities to stress or environmental changes was shown to be driven by the increase in abundance of some rare taxa (*Shade et al., 2014*; *Genitsaris et al., 2015*; *Coveley, Elshahed & Youssef, 2015*; *Gomez-Alvarez et al., 2016*).

A notable key point of our proposal is to estimate beta-diversity using $k$-mers diversity only. We are conscious this may lead to biased estimates of the beta-diversities defined from species composition data. The bias can run both ways: on the one hand, shared genomic regions or horizontal transfers between species will bias the $k$-mer-based distance downwards. On the other hand, genome size heterogeneity and $k$-mer composition variation along a microbe genome will bias the $k$-mer-based distance upwards. However, species composition based approaches are not feasible for large read sets from complex ecosystems (soil, seawater) due to the lack of good references and/or mapping scaling limitations. Moreover, our proposal has the advantage of being a *de novo* approach, unbiased by reference banks inconsistency and incompleteness. Finally, numerical experiments on the HMP datasets show that $k$-mer based and taxonomic distances are well correlated ($r > 0.8$ for $k \geq 21$) and consequently that Simka recovers the same biological structure as taxonomic studies do.

There is nevertheless room for improving Simka distances. For instance, recently, *Břinda, Sykulski & Kucherov (2015)* showed that spaced seeds can improve the $k$-mer-based metagenomic classification obtained with the popular tool Kraken (*Wood & Salzberg, 2014*). Spaced seeds can be seen as non-contiguous $k$-mers allowing therefore a certain number of mismatches when comparing them. Being less stringent when comparing $k$-mers could lead to more accurate distances, especially for viral metagenomic fractions which contain a lot of mutated sequences.

# ACKNOWLEDGEMENTS

Authors warmly thank, Olivier Jaillon and Thomas Vannier from Genoscope (CEA-IG-LAGE) and Stéphane Robin from INRA AgroParisTech for providing their technical, biological and statistical expertise, as well as feedback during the conception of this manuscript. We thank the GenOuest BioInformatics Platform that provided the computing resources necessary for benchmarking.

## Funding

This work was supported by the French ANR-14-CE23-0001 Hydrogen Project. The funders had no role in study design, data collection and analysis, decision to publish, or preparation of the manuscript.

## Grant Disclosures

The following grant information was disclosed by the authors:
French ANR-14-CE23-0001 Hydrogen Project.

## Competing Interests

The authors declare there are no competing interests.

## Author Contributions

- Gaëtan Benoit conceived and designed the experiments, performed the experiments, analyzed the data, contributed reagents/materials/analysis tools, wrote the paper, prepared figures and/or tables, performed the computation work, reviewed drafts of the paper.
- Pierre Peterlongo, Mahendra Mariadassou, Sophie Schbath, Dominique Lavenier and Claire Lemaitre conceived and designed the experiments, analyzed the data, contributed reagents/materials/analysis tools, wrote the paper, performed the computation work, reviewed drafts of the paper.
- Erwan Drezen contributed reagents/materials/analysis tools, wrote the paper, performed the computation work, reviewed drafts of the paper.

## Data Availability

Source code: https://github.com/GATB/simka.

## Supplemental Information

Supplemental information for this article can be found online at http://dx.doi.org/10.7717/peerj-cs.94#supplemental-information.

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
