# Peer review of "Multiple comparative metagenomics using multiset *k*-mer counting"

_PeerJ Computer Science, doi:10.7717/peerj-cs.94_

## Round 0.1 · original submission · Minor Revisions

Thank you for this! The reviewers generally supported publication, with a few caveats. If you can address these I will take another look at it quickly.

In particular, please address concerns about normalization, the strength of conclusions are larger and more complex data sets (the HMP is hardly representative of environmental data sets), discuss scalability and I/O, and please specifically address what considerations might be needed when analyzing high diversity and low coverage data sets.

·

Basic reporting

No comments.

Experimental design

In the supplementary figure 1, why the running time becomes constant after k=17? The running time should grow exponentially with the kmer size. My interpretation is that the authors only picked 1M reads from each 20 samples. As a result the total number of kmers is about 20M which is much smaller than 4^17. I think the plot is unfair.

About a related issue in the same figure, when the data set is fixed, why the disk usage grows with the kmer size? Is it due to the difficulty of compression when there are more distinct kmers? Even so, I am surprised to see that the curve did not flat out until k>=51.

Validity of the findings

I’m wondering whether the sample size has an impact on the correlation, i.e. some samples have much more reads than other samples. Though some of the distance has normalization factor, can it normalize the factor of the data set size?
For example, in the BrayCurtis distance calculation, if one data set is just repeats of the other data set 100 times, then the distance will be close 1, where the distance should be 0.

Additional comments

In this paper, the author implemented a highly-parallel program Simka that can count the kmers from many metagenomic samples while computing the ecological distance with additive property between the samples. The authors did a thorough analysis showing that analyzing kmers gives similar results when using other more complex analyzing methods. Thus Simka can be applied to analyze large-scale metagenomic experiments.

The framework of Simka is solid. It is quite scalable with respect to time and memory footprint. However, Simka heavily uses disk and is not scalable with respect to disk usage.

·

Basic reporting

In this manuscript, the authors reported the development of a method to compare metagenomic datasets based on k-mer counting. Not like some other tools, this tool - Simka can not only calculate the Bray-Curtis similarity, but also many other similarity metrics, which is nice. In this method, the k-mers abundance profiles across the metagenomic datasets are calculated. However taking advantage of the additive nature of computing some similarity metrics, the k-mers abundance profiles do not need to be stored and so is the huge k-mer count matrix, which reduces the disk usage. The authors demonstrated the benchmarking of Simka compared to other tools and compared the similarity measurement computed with Simka to that computed using other methods like sequences alignment and taxonomic profiling. Generally the manuscript was well written, with comprehensive description of the methods, data and pipeline for reproducibility. The software package repository is well organized on Github and it has good and clear documentation, which is very nice. There are some comments below about this manuscript though.

Experimental design

Main Comments

1. One of the most challenging problems in using k-mer counting to compare metagenomics datasets is how to deal with the k-mers from sequencing errors. As the authors mentioned in line 196, many of the k-mers with very low abundance come from sequencing errors. The solution of this method is filtering out those k-mers with abundance as 1, with those “solid k-mers” left. This works fine with metagenomics data set with higher coverage, as shown in the manuscript, with HMP samples as the testing dataset. It will be interesting to see how this method performs for other metagenomic datasets with lower coverage or higher diversity, like some environmental data sets. The IMG/M datasets used in COMMET paper and the Global Ocean Sampling datasets used in Comareads and Mash are two good candidates since in this manuscript, the authors compared the performance of Simka with COMMET and Mash. Also in line 475, the authors mentioned “Simka is able to capture such subtle signal raises hope of drawing new interesting biological insights from the data, in particular for those metagenomics project lacking good references (soil, seawater for instance).” and in line 528, “However, species composition based approaches are not feasible for large read sets from complex ecosystems (soil, seawater) due to the lack of good references and/or mapping scaling limitations. Moreover, our proposal has the advantage of being a de novo approach, unbiased by reference banks inconsistency and incompleteness.” It will be great if there are experiment results on those soil, seawater samples that can support these points.

Validity of the findings

2. In various parts in this manuscript, the authors mentioned that the solid k-mers filtering out does not affect results (line 376, 442, 489). This may be due to the high coverage of the HMP data sets. In the discussion in line 369-380, the authors mentioned that a small proportion (15%) of k-mers account for 95% of all base pairs of the whole dataset, which demonstrates that the HMP datasets have relatively higher sequencing coverage and most of the low abundance k-mers filtered out are probably sequencing errors. This may explain why the Simka results are robust with filtering (line 441) Just claiming that filtering out low abundance k-mers does not affect similarity measurement may not be accurate, at least before we see how this works for other environmental data sets with lower coverage recommended above. It will be nice if the authors can explain this more clearly.

3. Similarly, in line 490-493, the authors mentioned the filter can be disabled for samples with low coverage or where the rare species have more impact. But in this situation, how to deal with those large amount of erroneous k-mers? How will this affect the performance? In line 493-495, the authors claimed Simka is able to scale without solid k-mers filter. This may be true for the HMP data shown in the manuscript, but we still need to wait to see how it works for the low coverage data sets.

Additional comments

1. In Abstract- Methods, “Simka scales-up today metagenomic projects thanks to a new parallel k-mer counting strategy on multiple datasets.” Should “today” be “today’s”?
2. Line 3-6, “In large scale metagenomics studies such as Tara Oceans (Karsenti et al., 2011) or Human Microbiome Project (HMP) (Human Microbiome Project Consortium, 2012a) most of the sequenced data comes from unknown organisms and their short reads assembly remains an inaccessible task”. But in line 330, “One advantage of this dataset(HMP) is that it has been extensively studied, in particular the microbial communities are relatively well represented in reference databases” The descriptions about HMP seem like a contradiction here.
3. Table2, “2X16G paired reads”. it may be better to be just “2 X 16 billion paired reads”. I am not quite sure if “G” can be used like this.
4. Line 433, “On the other hand, Mash distances correlate badly with taxonomic ones (r = 0.51, see the comparison protocol in Article S1). “ It will be nice to cite Figure S3 here.
5. Line133 “For example, experiments on the HMP (Human Microbiome Project Consortium, 134 2012a) datasets (690 datasets containing on average 45 millions of reads each) require a storage space of 630TB for the matrix KC. “ How does the “630TB” calculated? For this method, the k-mer counting matrix and k-mer frequency vectors do not need to be stored. But the frequencies of all the k-mers in each partitions across difference data sets (red squares in Figure 2) are still stored on the disk after the sorting count stage, right? If so, how different is it compared to storing the k-mer counting matrix?
6. What is the role of the GATB library in Simka? If the GATB does the actual sorting count, then the paragraph in this manuscript about sorting count may be condensed. Also the description about the work of Chao et al. (2006) can be more precise too.
7. The software package repository is well organized and with good documentation. Just while I was trying to run the example test with “./bin/simka -in example/simka_input.txt -out results -out-tmp temp_output”, it failed with “Illegal instruction: 4”. I was using simka-v1.3.0-bin-Darwin.tar.gz and on mac OS 10.11. It may be the problem on my side. But this may be good for the authors to know.

---

## Round 0.2 · accepted · Accept

Thank you for making the requested revisions!

---

## Author Rebuttal · Round 0.2

Dear editors,

We are pleased to re-submit the revised version of our manuscript entitled "Multiple comparative metagenomics using multiset $k$-mer counting" by Benoit et al, for consideration for publication in Peerj journal. We would like to thank the editor and the reviewers for taking time and efforts to review our manuscript. You will find below a point by point response to the reviewers. Our responses are indicated in boxed text, and the corresponding parts that we have edited in the main document are indicated in red.

We thank you in advance for the attention you will give to our reply and we hope that the revised version of our manuscript will meet the criteria for publication in Peerj.

G. Benoit on the behalf of all authors.

———————————————

# Reviewer 1: Li Song

## Basic reporting

No comments.

## Experimental design

In the supplementary figure 1, why the running time becomes constant after k=17? The running time should grow exponentially with the kmer size. My interpretation is that the authors only picked 1M reads from each 20 samples. As a result the total number of kmers is about 20M which is much smaller than $4^{17}$. I think the plot is unfair.

> The 1M sub-sampling of the samples is not responsible for the performance behaviors presented in supplementary figure 1. To ensure this, we performed the same experiment with the full samples (no more sub-sampling) and we obtained similar behaviors. We changed the supplemental figure 1 accordingly. The fact that the running time does not grow exponentially with $k$ can be explained as follows. As explained lines 369-380 and Table 2, the running time of simka depends mainly on the number of solid distinct $k$-mers $W_s$ in the dataset. As the data are not random sequences and if $k$ is large enough (typically $k >= 17$), not all possible $k$-mers ($4^k$) can be seen in the data even if the samples contain an infinite number of reads. If $k$ is large enough (so that there are few repeated $k$-mers inside and between the genomes) and $k$-mers with sequencing errors are filtered out, $W_s$ is usually considered as the cumulated size of the genomes, therefore $W_s$ does not depend on $k$. This explains why the running time remains constant after $k = 17$.

About a related issue in the same figure, when the data set is fixed, why the disk usage grows with the kmer size? Is it due to the difficulty of compression when there are more distinct kmers? Even so, I am surprised to see that the curve did not flat out until $k >= 51$.

> Conversely, the disk usage depends on $k$. Even if the number of $k$-mers to store is constant with $k$, the space of each $k$-mer grows linearly with $k$ ($k$ characters to store).

## Validity of the findings

Im wondering whether the sample size has an impact on the correlation, i.e. some samples have much more reads than other samples. Though some of the distance has normalization factor, can it normalize the factor of the data set size? For example, in the BrayCurtis distance calculation, if one data set is just repeats of the other data set 100 times, then the distance will be close 1, where the distance should be 0.

> The normalization factors contained in the distance formula always concern a number of $k$-mers. Depending on the distance and on the parameters, this can be all the $k$-mers, only distinct ones or only solid distinct ones, and so on. In some datasets and cases, this may be linearly related to the data set size (for instance, in the given example, if one uses the presence-absence Jaccard index, the distance would be 0, as expected.), but in general this is surely not the case. We also remark that the shortcoming pointed out by the reviewer is inherent to the definition of the Bray-Curtis distance (and many others) rather than an artifact introduced by Simka. Therefore, if the sample sizes are significantly different between samples, we recommend, as is standard in community ecology, to use the option *-max-reads* of Simka to consider the same amount of reads for all datasets.

## Comments for the author

In this paper, the author implemented a highly-parallel program Simka that can count the kmers from many metagenomic samples while computing the ecological distance with additive property between the samples. The authors did a thorough analysis showing that analyzing kmers gives similar results when using other more complex analyzing methods. Thus Simka can be applied to analyze large-scale metagenomic experiments.

The framework of Simka is solid. It is quite scalable with respect to time and memory footprint. However, Simka heavily uses disk and is not scalable with respect to disk usage.

# Reviewer 2: Qingpeng Zhang

## Basic reporting

In this manuscript, the authors reported the development of a method to compare metagenomic datasets based on k-mer counting. Not like some other tools, this tool - Simka can not only calculate the Bray-Curtis similarity, but also many other similarity metrics, which is nice. In this method, the k-mers abundance profiles across the metagenomic datasets are calculated. However taking advantage of the additive nature of computing some similarity metrics, the k-mers abundance profiles do not need to be stored and so is the huge k-mer count matrix, which reduces the disk usage. The authors demonstrated the benchmarking of Simka compared to other tools and compared the similarity measurement computed with Simka to that computed using other methods like sequences alignment and taxonomic profiling. Generally the manuscript was well written, with comprehensive description of the methods, data and pipeline for reproducibility. The software package repository is well organized on Github and it has good and clear documentation, which is very nice. There are some comments below about this manuscript though.

## Experimental design

Main Comments

1. One of the most challenging problems in using k-mer counting to compare metagenomics datasets is how to deal with the k-mers from sequencing errors. As the authors mentioned in line 196, many of the k-mers with very low abundance come from sequencing errors. The solution of this method is filtering out those k-mers with abundance as 1, with those solid k-mers left. This works fine with metagenomics data set with higher coverage, as shown in the manuscript, with HMP samples as the testing dataset. It will be interesting to see how this method performs for other metagenomic datasets with lower coverage or higher diversity, like some environmental data sets. The IMG/M datasets used in COMMET paper and the Global Ocean Sampling datasets used in Comareads and Mash are two good candidates since in this manuscript, the authors compared the performance of Simka with COMMET and Mash. Also in line 475, the authors mentioned Simka is able to capture such subtle signal raises hope of drawing new interesting biological insights from the data, in particular for those metagenomics project lacking good references (soil, seawater for instance). and in line 528, However, species composition based approaches are not feasible for large read sets from complex ecosystems (soil, seawater) due to the lack of good references and/or mapping scaling limitations. Moreover, our proposal has the advantage of being a de novo approach, unbiased by reference banks inconsistency and incompleteness. It will be great if there are experiment results on those soil, seawater samples that can support these points.

> This is indeed a very interesting question. We agree that the HMP dataset is not representative of all metagenomic datasets, especially concerning the level of coverage of the genomes and that this feature may have strong impact on the solid kmer filter. Consequently, we added some experiments on the GOS dataset, as suggested by the reviewer. We first quantified the impact of the solid kmer filter on this dataset: we computed the correlation between distances obtained with or without this filter. Although it is lower than for the HMP gut dataset (0.999), the correlation obtained for the low-coverage GOS dataset is quite high: 0.97 (Spearman correlation on Bray-Curtis $k = 21$ distances). We also confirmed that Simka was able to recover the biological structure of the GOS samples: GOS samples are clustered according to their ocean origin (see heatmaps and sample classifications in Supplementary Figure 6), and that these qualitative results are robust with the use of the solid $k$-mer filter. These results are shown in details in the Supplementary file and are now discussed in a new paragraph in the Discussion section of the main manuscript.
>
> Even if, thanks to the reviewer remark, we added results on a sea water project, there is still large room for new applications over various heterogeneous projects. However, we reach here the limits of this technical paper, which aims to provide algorithmic description, biological validations as fair as possible, as well as discussions about possible future applications.

## Validity of the findings

2. In various parts in this manuscript, the authors mentioned that the solid k-mers filtering out does not affect results (line 376, 442, 489). This may be due to the high coverage of the HMP data sets. In the discussion in line 369-380, the authors mentioned that a small proportion (15%) of k-mers account for 95% of all base pairs of the whole dataset, which demonstrates that the HMP datasets have relatively higher sequencing coverage and most of the low abundance k-mers filtered out are probably sequencing errors. This may explain why the Simka results are robust with filtering (line 441) Just claiming that filtering out low abundance k-mers does not affect similarity measurement may not be accurate, at least before we see how this works for other environmental data sets with lower coverage recommended above. It will be nice if the authors can explain this more clearly.

> Again, we agree with the reviewer's point. The experiments we conducted on the GOS dataset show that results are also robust with the solid filter on this low-coverage dataset. As requested by the reviewer, we now clarify and discuss this point in the Discussion section.

3. Similarly, in line 490-493, the authors mentioned the filter can be disabled for samples with low coverage or where the rare species have more impact. But in this situation, how to deal with those large amount of erroneous k-mers? How will this affect the performance?

In line 493-495, the authors claimed Simka is able to scale without solid k-mers filter. This may be true for the HMP data shown in the manuscript, but we still need to wait to see how it works for the low coverage data sets.

> We have shown in the manuscript that the running time of Simka depends mainly on the number of distinct $k$-mers in the whole dataset. Indeed, when disabling the solid $k$-mer filter, the number of distinct $k$-mers can increase greatly. Whatever the coverage of the dataset, with the same number of reads, the number of sequencing errors and therefore the number of additional distinct $k$-mers due to sequencing errors will be roughly the same. Therefore, the impact of sequencing errors on Simka running time will be the same for high and low coverage datasets. What really matters is the total number of distinct $k$-mers, and that is the reason why we chose one of the largest publicly available dataset (HMP) to analyze Simka computational performances. We have shown that Simka has reasonable running time when dealing with hundreds of billions of distinct $k$-mers.

## Comments for the author

1. In Abstract- Methods, Simka scales-up today metagenomic projects thanks to a new parallel k-mer counting strategy on multiple datasets. Should today be todays?

> Typo corrected.

2. Line 3-6, In large scale metagenomics studies such as Tara Oceans (Karsenti et al., 2011) or Human Microbiome Project (HMP) (Human Microbiome Project Consortium, 2012a) most of the sequenced data comes from unknown organisms and their short reads assembly remains an inaccessible task. But in line 330, One advantage of this dataset(HMP) is that it has been extensively studied, in particular the microbial communities are relatively well represented in reference databases The descriptions about HMP seem like a contradiction here.

> We removed the reference to the HMP project in the first sentence (line 3-6).

3. Table2, 2X16G paired reads. it may be better to be just 2 X 16 billion paired reads. I am not quite sure if G can be used like this.

> We replaced "G" by "billion".

4. Line 433, On the other hand, Mash distances correlate badly with taxonomic ones (r = 0.51, see the comparison protocol in Article S1). It will be nice to cite Figure S3 here.

The citation has been added.

5. Line133 For example, experiments on the HMP (Human Microbiome Project Consortium, 134 2012a) datasets (690 datasets containing on average 45 millions of reads each) require a storage space of 630TB for the matrix KC. How does the 630TB calculated? For this method, the k-mer counting matrix and k-mer frequency vectors do not need to be stored. But the frequencies of all the k-mers in each partitions across difference data sets (red squares in Figure 2) are still stored on the disk after the sorting count stage, right? If so, how different is it compared to storing the k-mer counting matrix?

The reviewer raises an interesting point that needed to be clarified in the manuscript. This number was computed as follows: $W_s * (8 + 4N)$ bytes, with $W_s$ the number of distinct solid $k$-mers and $N$ the number of samples. The first 8 bytes are for storing each 31-mer, the $4N$ bytes are for storing the $N$ counts of each distinct $k$-mer. We used $W_s = 251 \times 10^9$ (line 4 of table 2), but this is a mistake, we should have used the number "after merging" $W_s = 95 \times 10^9$ (line 5 of table2) (in fact, $W_s = 251 \times 10^9$ may be considered as a worst case situation where samples share no $k$-mers). Thanks to the reviewer careful look, we therefore corrected this number in the manuscript from 630 TB to 260 TB, and explained how it is calculated. This does not impact the message of this paragraph. The example of the KC matrix storage space was intended to demonstrate that storing this matrix in main memory is not feasible, this is still the case with the correct number. We also clarified in the manuscript this detail (main memory), since main memory resources are much more limited than disk space. This is nevertheless interesting to compare it with the disk space used by the Simka partition files, in this case (1.6 TB, see table2). Apart from the compression share, we assume that the difference between the two numbers is mainly due to the fact that the matrix is very sparse, with lots of 0, and the absence of $k$-mers is not stored in the partition files.

6. What is the role of the GATB library in Simka? If the GATB does the actual sorting count, then the paragraph in this manuscript about sorting count may be condensed. Also the description about the work of Chao et al. (2006) can be more precise too.

Actually, the sorting count is implemented in the GATB library. However, some features of this sorting count, such as the multiple datasets feature and the merging count, were developed for Simka purpose and then integrated in the GATB library, and are not described in the older GATB publication. We therefore argue that this paragraph is important for the understanding of the whole method. Concerning the Chao et al. (2006) description, we added some details as suggested by the reviewer.

7. The software package repository is well organized and with good documentation. Just while I was trying to run the example test with `./bin/simka -in example/simka_input.txt -out results -out-tmp temp_output`, it failed with Illegal instruction: 4. I was using simka-v1.3.0-bin-Darwin.tar.gz and on mac OS 10.11. It may be the problem on my side. But this may be good for the authors to know.

> We are thankful to the reviewer for this feedback. However we were unable to reproduce the error.